# Comparative proteomics reveals *Cryptosporidium parvum* manipulation of the host cell molecular expression and immune response

Teng Li[1,2], Hua Liu[1], Nan Jiang[1], Yiluo Wang[1], Ying Wang[1], Jing Zhang[1], Yujuan Shen[1,2]*, Jianping Cao[1,2]*

**1** National Institute of Parasitic Diseases, Chinese Center for Disease Control and Prevention (Chinese Center for Tropical Diseases Research); Key Laboratory of Parasite and Vector Biology, National Health Commission of People's Republic of China; WHO Collaborating Center for Tropical Diseases, Shanghai, China, **2** The School of Global Health, Chinese Center for Tropical Diseases Research, Shanghai Jiao Tong University School of Medicine, Shanghai, China

* shenyj@nipd.chinacdc.cn (YS); caojp@yahoo.com (JC)

**Data Availability Statement:** The mass spectrometry proteomics data have been deposited at iProX with the data set identifier

## Abstract

*Cryptosporidium* is a life-threating protozoan parasite belonging to the phylum Apicomplexa, which mainly causes gastroenteritis in a variety of vertebrate hosts. Currently, there is a re-emergence of *Cryptosporidium* infection; however, no fully effective drug or vaccine is available to treat Cryptosporidiosis. In the present study, to better understand the detailed interaction between the host and *Cryptosporidium parvum*, a large-scale label-free proteomics study was conducted to characterize the changes to the proteome induced by *C. parvum* infection. Among 4406 proteins identified, 121 proteins were identified as differentially abundant (> 1.5-fold cutoff, $P < 0.05$) in *C. parvum* infected HCT-8 cells compared with uninfected cells. Among them, 67 proteins were upregulated, and 54 proteins were downregulated at 36 h post infection. Analysis of the differentially abundant proteins revealed an interferon-centered immune response of the host cells against *C. parvum* infection and extensive inhibition of metabolism-related enzymes in the host cells caused by infection. Several proteins were further verified using quantitative real-time reverse transcription polymerase chain reaction and western blotting. This systematic analysis of the proteomics of *C. parvum*-infected HCT-8 cells identified a wide range of functional proteins that participate in host anti-parasite immunity or act as potential targets during infection, providing new insights into the molecular mechanism of *C. parvum* infection.

## Author summary

*Cryptosporidium parvum* is an emerging zoonotic pathogen transmitted via the fecal–oral route, and is considered a leading cause of moderate-to-severe diarrheal disease in young children in resource limited areas. After infection, *C. parvum* parasitizes intestinal epithelial cells and evokes an inflammatory immune response, leading to severe damage of the

IPX0003333000, subproject IPX0003333001.
https://www.iprox.cn/page/project.html?id=
IPX0003333000.

**Funding:** This work was supported by the National
Key R&D Program of China (No. 2017YFD0500400
to HL), the National Natural Science Foundation of
China (Nos. 81772225 and 81971969 to JC), the
National Science and Technology Major Project
(No. 2018ZX10713001-004 to YS) and the Fifth
Round of Three-Year Public Health Action Plan of
Shanghai, China (No. GWV-10.1-XK13 to JC). The
funders had no role in study design, data collection
and analysis, decision to publish, or preparation of
the manuscript.

**Competing interests:** The authors have declared
that no competing interests exist.

intestinal mucosa. The infection can be lethal to immunosuppressed individuals. How-
ever, no fully effective drug or vaccine is available for cryptosporidiosis, and the pathogen-
esis and immune mechanisms during *C. parvum* infection are obscure. Thus, an in-depth
understanding of host-parasite interaction is needed. Hence, we established a *C. parvum*-
infected HCT-8 cell model and performed comparative quantitative proteomic analyses
to profile global host-parasite interactions and determine the molecular mechanisms that
are activated during infection, aiming to offer new insights into the treatment of
Cryptosporidium.

## 1. Introduction

*Cryptosporidium* is an opportunistic pathogen with a worldwide distribution, which infects a
variety of vertebrates (including humans, mammals, reptiles, amphibians, and poultry) [1].
Immunocompetent individuals experience a self-limiting illness after *C. parvum* infection;
immunocompromised hosts suffer from more severe and prolonged gastrointestinal disease
that can be fatal [2,3]. Nearly 40 genotypes of *Cryptosporidium* have been established, and it is
likely that after further biological and molecular characterization, many of these will eventually
be given species status [4].

   *Cryptosporidium* parasitizes mainly the epithelial cells of the gastrointestinal and respiratory
tracts, evoking host epithelial defense responses mediated by Toll like receptors (TLRs) [5–8].
In contrast to other apicomplexans, such as *Toxoplasma gondii* and *Plasmodium falciparum*,
*Cryptosporidium* has lost the plastid and mitochondrial genomes, and both the asexual and
sexual stages are completed within a single host [9–11]. As a result, *Cryptosporidium* metabo-
lism is almost exclusively based on glycolysis, which is likely to be the reason why it parasitizes
mainly gut epithelial cells [9,10]. Interestingly, *C. parvum* does not fully invade the host cell,
but resides intracellularly in the parasitophorous vacuole with an epicellular location.

   *Cryptosporidium* forms an actin-rich disk as a feeder organelle, which is an important strat-
egy for successful nutritional uptake and rapid replication [12–16]. However, the molecular
mechanisms in the host and parasite that lead to this epicellular niche remain unknown. More-
over, evidence shows that some host-parasite homologous genes of the host glycolysis/gluco-
neogenesis pathways are downregulated, while host-exclusive genes are upregulated during
invasion and intracellular development. This might suggest parasite-derived competition for
metabolic substrates [17]. Although evidence shows that multiple immune cells and effectors
participate in resistance to *C. parvum* infection or parasite clearage, infected cells eventually
die because of nutrient deprivation and disruption of the cytoskeleton [18–20]. Our knowledge
of the molecular mechanisms of *C. parvum*-host interactions is limited and no effective anti-
cryptosporidial therapies are available to treat cryptosporidiosis in children, patients with
HIV/AIDS, and animals [21–25]. Thus, there is an urgent need to gain an in-depth under-
standing of host-parasite interactions and develop effective drugs and vaccines.

   In the present study, we aimed to use a highly sensitive quantitative approach, combining
label-free proteomic quantification techniques on a liquid chromatography-dual mass spec-
trometry (LC-MS/MS) platform with advanced bioinformatic analysis, to analyze the global
proteome differences in *C. parvum*-infected and non-infected HCT-8 cells. Moreover, the
authenticity and accuracy of the protein levels identified by the quantitative proteomic exami-
nation were further confirmed using western blotting and quantitative real-time reverse tran-
scription polymerase chain reaction (qRT-PCR). This study aimed to identify molecules that
potentially play important roles during *C. parvum* infection.

## 2. Materials and methods

### 2.1 Sample preparation and collection

*C. parvum* oocysts of the Iowa strain were purchased from a commercial source (Waterborne, New Orleans, LA, USA). The HCT-8 human colon adenocarcinoma cell line (ATCC, Manassas, VA, USA) was cultured in Dulbecco's modified Eagle's medium (DMEM) nutrient mix F-12 (Gibco, Grand Island, NY, USA) containing 10% fetal bovine serum (FBS, Gibco). For infection, *C. parvum* oocysts were suspended in phosphate-buffered saline (PBS) with 20% sodium hypochlorite, centrifuged at $3,667 \times g$, and washed twice with PBS. Infection was performed in culture medium containing viable *C. parvum* oocysts (at an oocyst to host cell ratio of 2:1) at 37°C in a humidified 5% $CO_2$ incubator for 36 h. An equal volume of culture medium was added to the control group. After 36 h of incubation, the cells were washed three times with cold PBS and collected using a cell scraper. The cells were centrifuged at $200 \times g$ for 10 min to obtain the cell pellet, which was stored at -80°C until analysis.

### 2.2 Quantitative real-time reverse transcription PCR (qRT-PCR)

Total RNA from the HCT-8 cells was extracted using the TRIzol reagent according to the manufacturer's protocol (Takara Biotechnology, Inc., Shiga, Japan). The primers were designed by EnzyArtisan Biotech Co., Ltd. (Shanghai, China, Table 1). The cDNA was prepared using a PrimeScript RT–PCR Kit (Takara Biotechnology, Inc.) and then amplified using a SYBR qPCR Mix (EnzyArtisan Biotech). The *ACTB* (β-actin) gene was used as an endogenous control for normalization.

### 2.3 Protein extraction and trypsin digestion

More than $2 \times 10^7$ cells of each group were collected and sonicated three times on ice using a high intensity ultrasonic processor (Scientz, Zhejiang, China) in lysis buffer (8 M urea, 1% Protease Inhibitor Cocktail). The lysate was centrifuged at $12,000 \times g$ at 4°C for 10 min to separate

**Table 1.  List of primers used for qRT-PCR analysis.**

| Gene | Primer sequence |
|------|-----------------|
| *ISG15* | Forward: CTCTGAGCATCCTGGTGAGGAA<br>Reverse: AAGGTCAGCCAGAACAGGTCGT |
| *IFITM3* | Forward: GTGCTGATCTTCCAGGCCTATG<br>Reverse: TGGAGTACGTGGGATACAGGTCAT |
| *PLSCR1* | Forward: ATTAAGAACAGCTTTGGACAGAGG<br>Reverse: TCCTCAAGGTAAAAGGTCTAGATGG |
| *NMI* | Forward: GAAACGGAGTTACAAGAGGCTAC<br>Reverse: GACAACTGGCTGTCATTCTCAGG |
| *SFPQ* | Forward:CTGTGTCATCCGCCATTTTGTG<br>Reverse: GGAACCGATCCCGAGACATG |
| *RBMX* | Forward: CACCTCGAAGGGAACCGCTG<br>Reverse: TCGTGGTGGTGGTGCATAATCTCTA |
| *IFI35* | Forward: CACGATCAACATGGAGGAGTGC<br>Reverse: GGCAGGAAATCCAGTGACCAAC |
| *HELZ2* | Forward: GAGGTGCATCTGTGTCGTTTCC<br>Reverse: CAGGATCTCAAAACTGCCGACAG |
| *ATP5PO* | Forward: GCCTAAATGACATCACAGCAAAAG<br>Reverse: AGGCAGAAACGACTCCTTGGGT |
| *ACTB* | Forward: CACCATTGGCAATGAGCGGTTC<br>Reverse: AGGTCTTTGCGGATGTCCACGT |

the insoluble debris. Finally, the soluble supernatant was collected and the protein concentration was determined using a bicinchoninic acid (BCA) kit according to the manufacturer's instructions (Beyotime, Jiangsu, China). Then, the protein supernatant was thoroughly digested using trypsin for further analysis.

## 2.4 LC-MS analysis and database searching

The tryptic peptides were loaded onto a laboratory-prepared reversed-phase analytical column (C18), subjected to a nanospray ionization (NSI) source followed by tandem mass spectrometry (MS/MS) in a Q Exactive Plus apparatus (ThermoScientific, Waltham, MA, USA) coupled online to an ultra-performance liquid chromatography (UPLC) column, and then selected for MS/MS using a normalized collision energy (NCE) setting of 28. The fragments were detected in the Orbitrap at a resolution of 17, 500. Automatic gain control (AGC) was set at 5e4 and the fixed first mass was set as 100 m/z. Then, the Maxquant search engine (v.1.5.2.8, Max Planck Institute of Biochemistry, Munich, German) was used for secondary mass spectral data process. All tandem mass (MS/MS) spectra were searched against human and *C. parvum* data in the Uniprot database, which was concatenated with a reverse decoy database to calculate the false discovery rate (FDR) caused by random matching. Trypsin/P was specified as the cleavage enzyme allowing up to two missing cleavages and the minimum length of the peptide segment was set as seven amino acid residues. 20 ppm and 5 ppm were set as the mass error tolerance of primary parent ions for the first and main search, respectively, and the mass error tolerance was set as 0.02 Da for secondary fragment ions.

## 2.5 Bioinformatic analysis

The UniProt-GOA database (http://www.ebi.ac.uk/GOA/), InterPro domain database (http://www.ebi.ac.uk/interpro/), and the Kyoto Encyclopedia of Genes and Genomes (KEGG) database (http://www.genome.jp/kaas-bin/kaas_main; http://www.kegg.jp/kegg/mapper.html) were used separately for Gene Ontology (GO) annotation, characterization of the proteins basic functions, domain functional descriptions, and annotation of biological processes or pathways. Then, a two-tailed Fisher's exact test was employed to test the enrichment of the differentially expressed proteins (DEPs) against all identified proteins. A *P*-value < 0.05 was considered significant. All the categories obtained after enrichment were collected and then filtered for those categories that were at least enriched in one of the clusters with a *P*-value < 0.05. This filtered *P*-value matrix was transformed and then clustered using one-way hierarchical clustering in Genesis. Clusters were visualized using the R Package, pheatmap (https://cran.r-project.org/web/packages/cluster/).

We used Wolfpsort (http://www.genscript.com/psort/wolf_psort.html) to predict the subcellular locations of the proteins. COG (Clusters of Orthologous Groups of proteins)/KOG (euKaryotic Ortholog Groups) functional classification of the identified proteins was conducted through database comparison and analysis.

## 2.6 Protein-protein interaction network

All DEPs' database accession numbers or sequences were searched against STRING database version 11[26] for protein-protein interaction (PPI) analysis. Only interactions between the proteins belonging to the searched data set were selected, thereby excluding external candidates. The interaction network from STRING was visualized in Cytoscape [27].

## 2.7 Western blotting

*C. parvum* infected and non-infected HCT-8 cells were incubated for 36 h before harvesting. Equal amounts of protein from total-cell lysates were separated using 12% SDS-PAGE (Beyotime) and transferred onto polyvinylidene fluoride (PVDF) membranes (Millipore, Billerica, MA, USA). The membranes were then blocked at room temperature for 2.5 h in 5% nonfat dry milk in Tris-buffered saline-Tween 20 (TBS-T) buffer. Membranes were incubated with gentle rocking 1.5 h at room temperature with primary antibodies recognizing interferon-induced 15 KDa protein (ISG15) (1/5000, Abcam, Cambridge, MA, USA), interferon induced transmembrane protein 3 (IFITM3) (1/5000, Abcam), phospholipid scramblase 1 (PLSCR1) (1/5000, Abcam), N-Myc and STAT interactor (NMI) (1/1000, Proteintech, Wuhan, China), signal transducer and activator of transcription (STAT1) (1/2000, Proteintech), STAT2 (1/2000, Abcam), splicing factor proline and glutamine rich (SFPQ) (1/500, Abcam), RNA binding motif protein X-linked (RBMX) (1/5000, Abcam), S100 calcium binding protein A10 (S100A10) (1/2000, Abcam), and β-actin (1/20000, Proteintech) as a loading control. Membranes were washed with TBS-T five times for 10 min each, and then incubated with the appropriate secondary antibody for 1 h at room temperature. The immune complexes were then visualized using an enhanced chemiluminescence (ECL) detection kit (Tanon, Shanghai, China). The densities of the immunoreactive protein bands were determined using Image J software (NIH, Bethesda, MD, USA).

## 2.8 Statistical analysis

Data are presented as means ± SD. Statistical tests were performed using the GraphPad Prism 8 Statistics software package (GraphPad Inc., La Jolla, CA, USA). A value of $P < 0.05$ was considered statistically significant.

# 3. Results

## 3.1 Proteomic analysis of DEPs in non-infected and infected samples

A total of 4406 human proteins were identified, of which 3307 contained quantitative information, ranging in molecular weight from 1.9 kDa to 964.8 kDa (S1 Fig). Among them, 33, 812 acquired unique peptides belonged to 3307 proteins, which served as reliable evidence for these predicted proteins (S1 Fig). Most of the identified proteins contained no less than two unique peptides as multiple peptide-level evidence for protein identification (S1 Fig). By searching the database of *C. parvum*, we obtained 231 proteins and 563 unique peptides, indicating that a considerable number of proteins are single peptide chains (S2 Fig).

## 3.2 Bioinformatic analysis of host proteomic data

### 3.2.1 Subcellular localization of DEPs.
A mass spectrometry comparison between infected and non-infected HCT-8 cells identified 121 DEPs (1.5-fold difference in abundance), of which 64 were upregulated and 57 were downregulated after infection (Fig 1A and 1B). Interestingly, among the DEPs, those located on the plasma membrane were all upregulated after infection (S1 Table). Besides, the upregulated and downregulated proteins had various locations in the cellular compartments, including the cytoplasm, nucleus, mitochondria, and extracellular region (Fig 1B). In-depth analysis further demonstrated the different locations of DEPs caused by infection. Functional enrichment analysis indicated that the upregulated proteins were concentrated in the transporters associated with antigen processing (TAP) complex, extracellular space, intrinsic component of plasma membrane, and MHC class I peptide loading complex. The downregulated proteins were found to be mostly located in the organelle

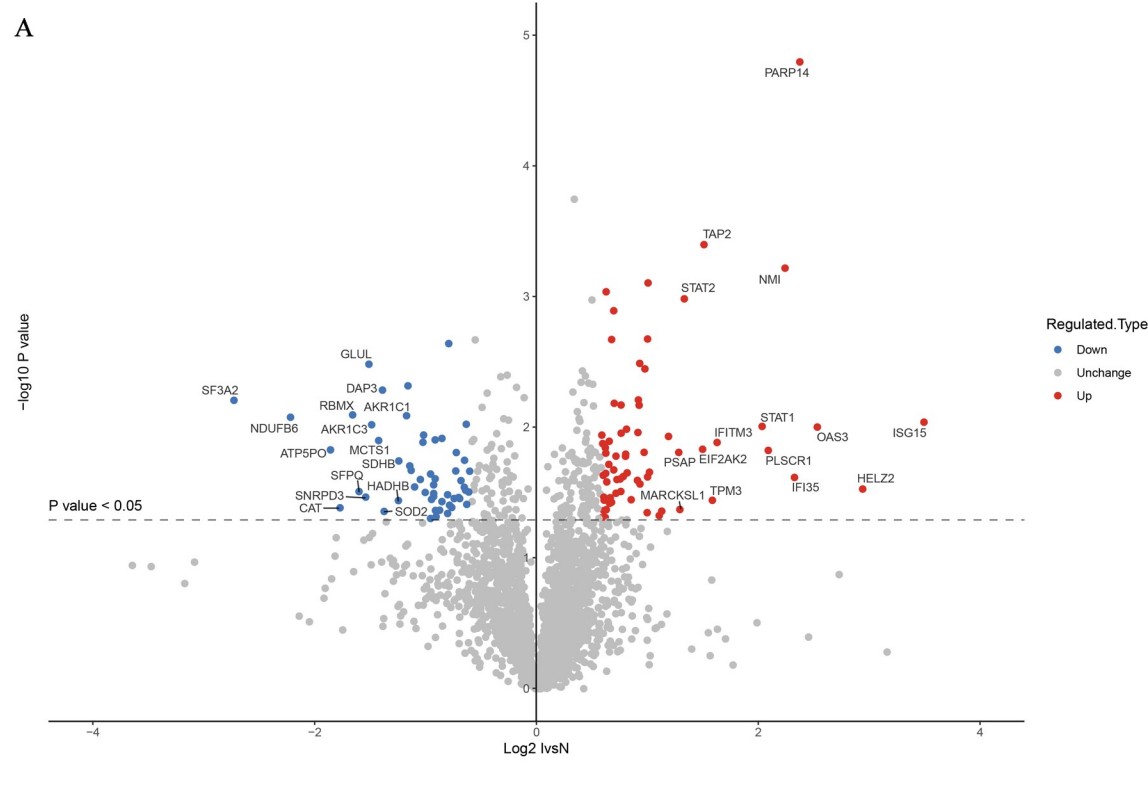

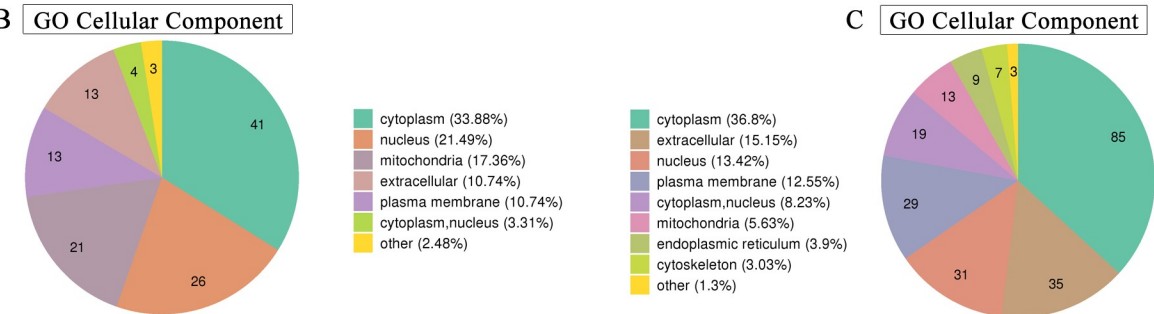

**Fig 1. *C. parvum* infection altered the levels of multiple proteins in host cells. (A)** Volcano plot of the most significantly DEPs between *C. parvum* infected and uninfected HCT-8 cells. The horizontal axis shows the relative quantitative value of protein levels after $Log_2$ logarithmic conversion, the vertical axis show the *P* values of significant difference test after $Log_{10}$ logarithmic conversion. In this figure, red dots represent proteins with significantly upregulated levels, while blue dots represent downregulated proteins. **(B)** Subcellular localization and distribution of DEPs in HCT-8 cells. **(C)** Location and classification of subcellular structures analysis of identified *C. parvum* proteins.

lumen, membrane-enclosed lumen, intracellular organelle lumen, microbody lumen, and peroxisomal matrix (S3 Fig). The obvious local distinction between upregulated and downregulated proteins suggested the special effect of *C. parvum* infection on various components of the cell.

In addition, we conducted bioinformatic analysis on the intracellular parasite components, and 231 *C. parvum*-derived proteins were identified at 36 h post infection (Figs 1C and S4). These parasitic proteins were enriched significantly in the cytoplasm, extracellular space, nucleus, plasma membrane, nucleus, mitochondria, and endoplasmic reticulum (Fig 1C), which constitute the core components of *C. parvum*.

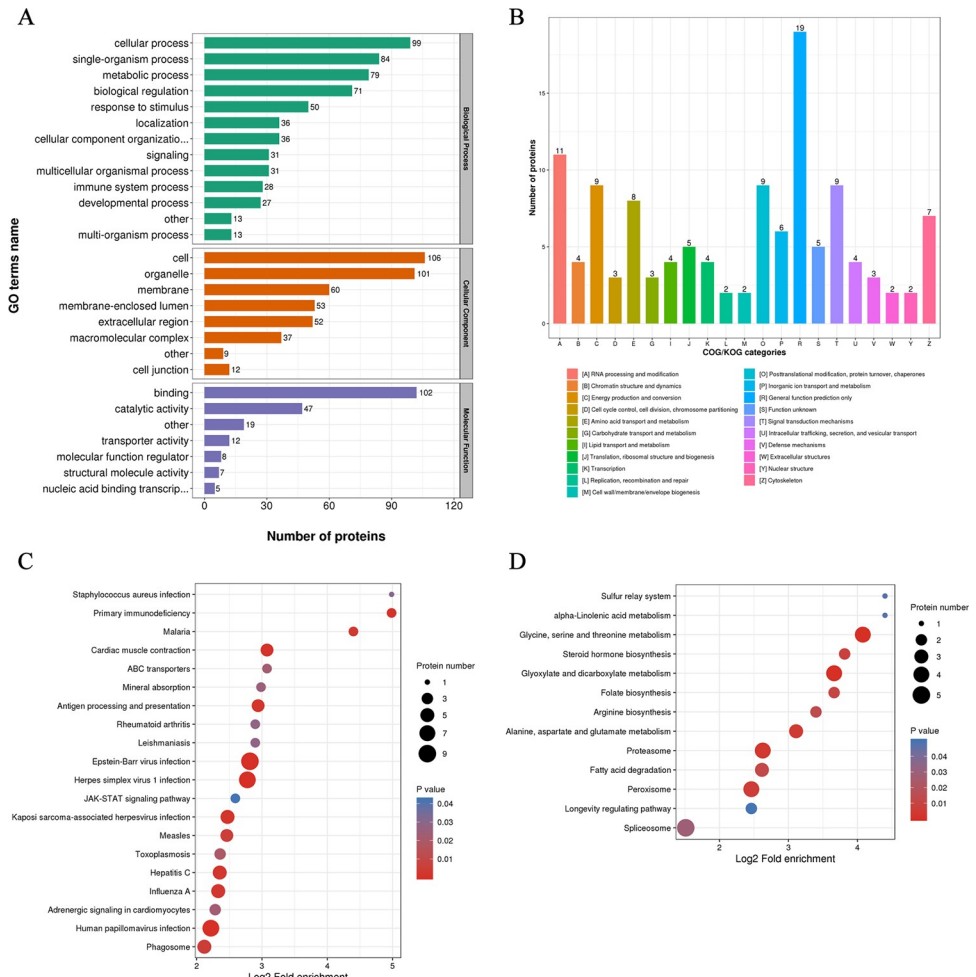

**Fig 2. Bioinformatic characterization of the DEPs in host cells.** **(A)** GO classification of DEPs according to their associated biological process, cellular component, and molecular function terms. **(B)** COG/KOG functional classification of DEPs. **(C)** KEGG pathways enrichment analysis of the upregulated proteins. **(D)** KEGG pathways enrichment analysis of the downregulated proteins. The color represents the significance *P*-value of enrichment. The size of the circle represents the number of proteins involved.

**3.2.2 GO enrichment and KEGG pathway enrichment of DEPs.** We further annotated and investigated the functions of the identified DEPs, with the aim of discovering important molecules or signaling pathways that play key roles during *C. parvum* parasitism. According to biological process analysis from the GO classification, the DEPs were closely involved in cellular process, single-organism process, metabolic process, response to stimulus, and biological regulation (Fig 2A). Among the 121 DEPs, 102 were identified as having binding functions and 47 had catalytic activity (Fig 2A). COG/KOG categories provided detailed functional classification of DEPs, suggesting that the *C. parvum* infection has extensive effects on various physiological activities of host cells, such as RNA processing and modification, signal transduction mechanisms, posttranslational modification, protein turnover, chaperones, energy production and conversion, amino acid transport, and metabolism (Fig 2B).

Based on further GO enrichment analysis, *C. parvum* infection seemed to evoke a series of cell defense reactions, in that the upregulated proteins were related to cellular response to type I interferon, and the defense response to virus and other organisms. By contrast, the

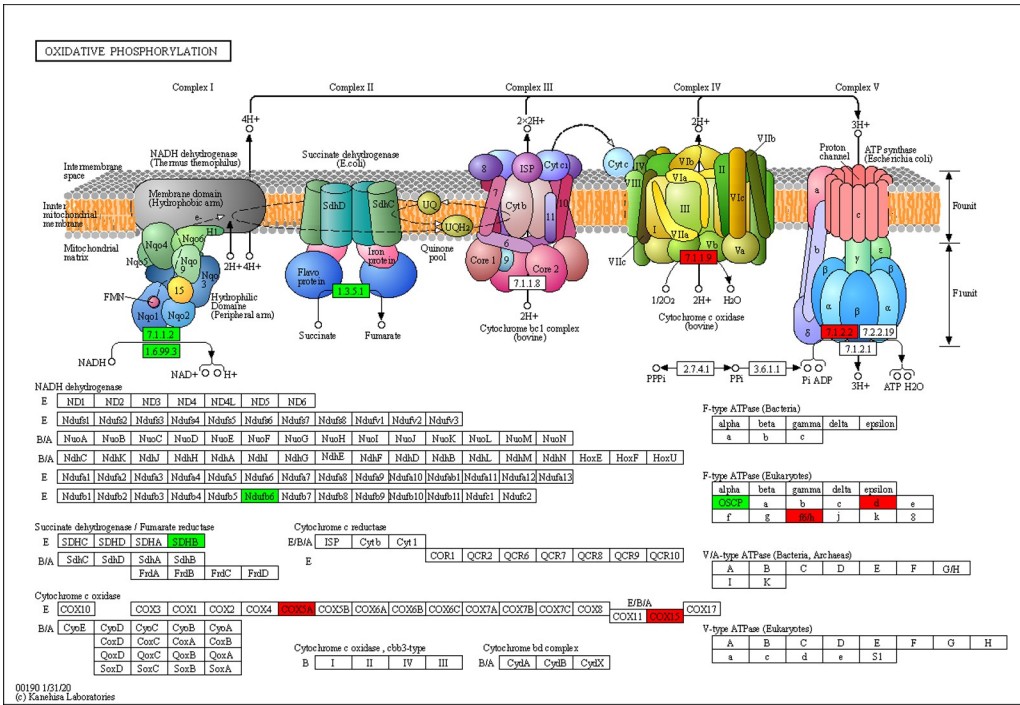

**Fig 3. *C. parvum* affected the oxidative phosphorylation process of host cells.** The protein names in red showed upregulated levels and those in green showed downregulated levels in *C. parvum*-infected HCT-8 cells compared with uninfected cells.

downregulated proteins were functionally enriched for antibiotic metabolic processes (including aminoglycoside antibiotic and doxorubicin metabolic process), positive regulation of inclusion body assembly, protein oligomerization, and hypoxia response due to infection. Moreover, several signaling pathways were affected by *C. parvum* infection. KEGG pathway enrichment analysis of upregulated and downregulated proteins was conducted separately, which indicated an obvious distinction in functional clusters between the DEPs (Fig 2C and 2D). Remarkably, the upregulated proteins exhibited significant enrichment for inflammatory signaling pathways, and were involved in many pathogenic pathways of virus and bacterial infections, Malaria, Leishmaniasis, and Toxoplasmosis (Fig 2C). These findings suggested that in addition to some specific pathways, different parasites or viruses might target common pathways and machineries, which forms the basis for the discovery of broad range vaccines or drugs. By contrast, the downregulated proteins caused by *C. parvum* infection were found to be markedly enriched in multiple metabolic pathways, including metabolism of many amino acids and propanoate, biosynthesis of folate, unsaturated fatty acids and steroid hormone, and the peroxisome pathway (Fig 2D). Besides, *C. parvum* had a significant effect on the host oxidative phosphorylation process, with up to 12 related proteins being differentially abundant after infection (Fig 3). The extensive influence of *C. parvum* on host metabolic process revealed *C. parvum*'s need for a special cellular microenvironment, and also reflected its dependence on certain nutrients, which provided ideas for identifying effective drugs or inhibitors.

### 3.3 Bioinformatic analysis of *C. parvum* proteomic data

To gain a deeper understanding of the biological characteristics of *C. parvum*, we also conducted a bioinformatic analysis on the identified *C. parvum*-derived proteins. The COG/KOG

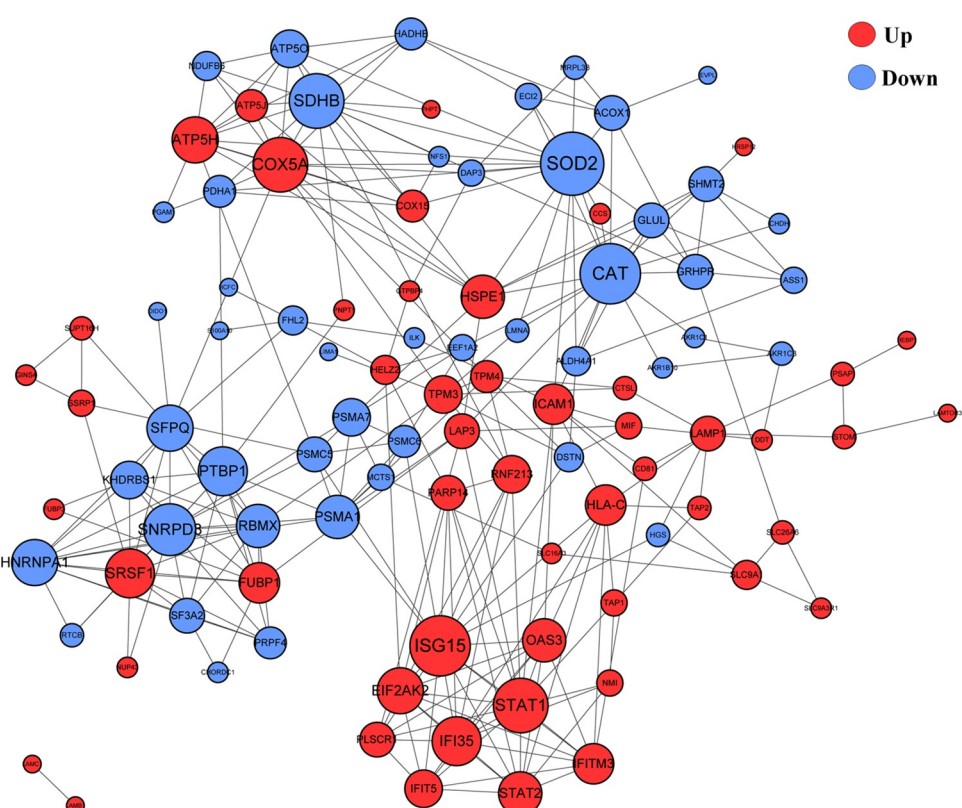

**Fig 4. Association networks of DEPs using Cytoscape.** PPI network profiles the interplay of the identified host proteins. The color of the circle represents upregulated proteins (red) and downregulated proteins (blue) respectively. The size of the circle represents the significance of the protein.

annotation revealed the predominant involvement of identified *C. parvum*-derived proteins in translation, ribosomal structure and biogenesis, posttranslational modification, protein turnover, and signal transduction processes (S4 Fig). During parasitism by *C. parvum*, identified proteins were enriched in the myelin sheath, lipid particle, and ATPase complex and were functionally specialized in the metabolic process of ATP, nicotinamide nucleotide, and ribonucleoside triphosphate (S4 Fig). KEGG pathway enrichment further illustrated a wide distribution of identified *C. parvum*-derived proteins in the ribosome after infection, which could be important for successful parasitism and replication (S4 Fig).

## 3.4 The DEPs exhibited close interactions in protein-protein interaction network

The variety and diverse functions of the identified proteins prompted us to construct a PPI network to better characterize the potential interaction of the host and *C. parvum* proteins. In total, 100 DEPs were mapped to a PPI network, comprising a dense interaction network of host proteins. Noticeably, several sub-networks with strong interactions were identified (Fig 4). The significant hub proteins included interferon stimulated gene 15 (ISG15), signal transducer and activator of transcription 1 (STAT1), cytochrome C oxidase subunit 5A (COX5A), succinate dehydrogenase (SDHB), superoxide dismutase 2 (SOD2), catalase (CAT), small nuclear ribonucleoprotein D3 polypeptide (SNRPD3), heat shock protein family E member 1 (HSPE1), proteasome subunit alpha type 1 (PMSA1), and polypyrimidine tract-binding protein 1 (PTBP1). All the hub proteins have potential roles in host-*C. parvum* interplay;

however, the detailed mechanisms require further study. Besides, 109 *C. parvum*-derived proteins were cross referenced with the STRING and Cytoscape databases for protein interaction analysis (S5 Fig). Thirty-two proteins were determined and clustered with a high degree of network interaction, and the hub proteins were identified, which included cgd7_2280, elongation factor 2, and cgd3_2090, identifying a series of parasitic proteins of vital importance, which might represent the core functional proteins of *C. parvum*.

## 3.5 Validation of the proteomic results

To assess the proteins that were identified as being closely related to the host anti-parasite mechanism, or associated with parasite development and pathogenicity during *C. parvum* infection, we verified their expression using qRT-PCR and western blotting. Firstly, the mRNA expression level of nine genes, including *ISG15*, *IFITM3* (interferon-induced transmembrane protein 3), *PLSCR1* (phospholipid scramblase 1), *NMI* (N-Myc and STAT interactor), *IFI35* (interferon-induced protein 35), *HELZ2* (helicase with zinc finger 2), *SFPQ* (splicing factor, proline- and glutamine-rich), *RBMX* (RNA-binding motif protein), and *ATP5PO* (ATP synthase peripheral stalk subunit OSCP) were compared in normal and infected HCT-8 cells, and the results were in accordance with the proteomic analysis (Fig 5A).

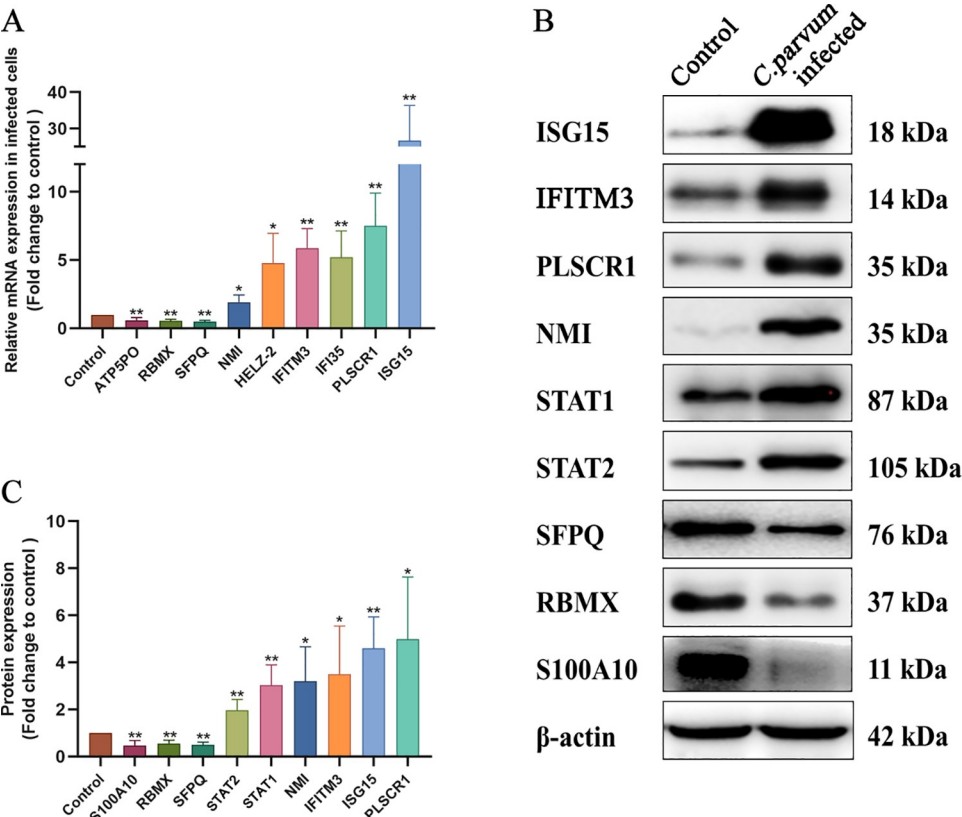

**Fig 5. Verification of the DEPs in normal and *C. parvum*-infected host cells.** Each experiment was repeated three times. **(A)** Relative mRNA expression levels of *ATP5PO*, *RBMX*, *SFPQ*, *NMI*, *HELZ2*, *IFITM3*, *IFI35*, *PLSCR1*, and *ISG15* in normal and *C. parvum*-infected HCT-8 cells were verified using quantitative real-time reverse transcription PCR. The error bars represent the mean ± SD, from one experiment run in triplicate. Statistical significance was analyzed using an unpaired Student's t-test (*$P < 0.05$, **$P < 0.01$). **(B)** Western blotting analysis was used to reconfirm the levels of the identified proteins with biological significance in infected and uninfected HCT-8 cells. **(C)** Histogram analysis of the western blotting results in infected and uninfected HCT-8 cells. Statistical significance was analyzed using an unpaired Student's t-test (*$P < 0.05$, ** $P < 0.01$).

Then, several proteins were chosen and further confirmed using western blotting, which showed that the levels of ISG15, IFITM3, PLSCR1, NMI, STAT1, signal transducer and activator of transcription 2 (STAT2), SFPQ, RBMX, and protein S100-A10 (S100A10) accorded with those in the proteomic analysis (Fig 5B and 5C).

## 4. Discussion

As an enteric pathogen, *C. parvum* primarily infects the apical end of small intestinal enterocytes, where they are enveloped by host membranes but remain extracytoplasmic [28,29]. Intestinal epithelial cells (IECs) can sense and participate in the coordination of appropriate immune responses, ranging from tolerance to anti-pathogen immunity [30,31]. Hence, IECs are considered as the frontline sensors for pathogenic microorganism encounters [32,33]. The HCT-8 cell line possesses an excellent ability to support *C. parvum* growth after infection [34–36]. In this study, we established a *C. parvum* infection model using HCT-8 cells to mimic the pathogenesis and immune reaction during *C. parvum* invasion and development, aiming to provide key insights into how the host reacts effectively to infection. Our data revealed that the levels of 121 host proteins were altered during *C. parvum* infection, which possessed diverse functions, and profiled various biological process changes in infected host cells globally.

Based on bioinformatic analysis and recent research evidence, we focused on several significant signaling pathways and molecules that might serve as important targets during the anti-*C. parvum* immune response. GO analysis revealed that the interferon (IFN) signaling pathway was the most enriched biological process, in that seven (10%) of the 67 significantly upregulated proteins were identified as targets of IFN signaling; specifically, ISG15, 2'-5'-oligoadenylate synthetase 3 (OAS3), NMI, IFITM3, STAT1, STAT2, and IFI35. Interferon-induced protein ISG15 has been implicated as a core player during the host antiviral response by directly disrupting viral replication, budding, and release, as well as by limiting tissue damage and modulating human type I interferon signaling [37–43]. The biological relevance of ISG15 in parasitic infection has been demonstrated. Novel findings suggest that ISG15 expression was induced during *Theileria annulata* and *Leishmania brazilensis* infections, but not in *L. amazonensis* infection [44,45]. Recently, ISG15 was found to participate in autophagy-mediated ubiquitinoylation of the vacuole during *T. gondii* infection, as a bridge that links the ATG pathway with IFN-γ-dependent restriction of *T. gondii* in host cells [46]. In the present study, significantly increased ISG15 expression was confirmed, indicating a potential anti-parasitic function of ISG15 during *C. parvum* infection. The PPI network further exhibited strong interaction among ISG15 and other upregulated proteins including OAS3, IFITM3, IFI35, intercellular adhesion molecule-1 (ICAM-1), STAT1, STAT2, PLSCR1, eukaryotic translation initiation factor 2 alpha kinase 2 (EIF2AK2), and NMI, most of which are interferon-induced proteins. The upregulation of IFITM3, STAT1, STAT2, PLSCR1, and NMI after infection was validated using western blotting and qRT-PCR. Based on these results, we identified that the prevailing immune response of HCT-8 cells to *C. parvum* is centered on IFN signaling, revealing the involvement of unique proteins and a distinct immune process in *C. parvum* infection, which might be key regulators in the anti-*C. parvum* immune reaction.

ICAM-1 is a cell surface glycoprotein and an adhesion receptor that is expressed widely by endothelial, epithelial, and immune cells, and is responsible for leukocyte recruitment and signal transduction [47]. Under inflammatory conditions or stimulation by injury, the expression of ICAM-1 markedly increased in immune cells and epithelial cells [48–50]. Recently, ICAM-1 has also been proven to be an important cytoadherence receptor for *Plasmodium chabaudi* infection, and binds to *Plasmodium falciparum* erythrocyte membrane protein 1, suggesting its important role in parasitic protozoa infection [51,52]. Furthermore, ICAM-1 expression

was induced in the rat jejunum during acute *T. gondii* infection [53]. Moreover, Chen *et al.* demonstrated that the upregulation of ICAM-1 in epithelial cells was modulated by micro-RNA-221 following *C. parvum* infection. ICAM-1 also enhanced the attachment of infected host cells [54]. In the present study, the robustly enhanced expression of ICAM-1 in *C. parvum* infected cells suggested that ICAM-1 might serve as a biosensor in the response to *C. parvum* infection and participate in the recruitment of immune cells. However, the detailed mechanism requires in-depth investigation.

In the present study, we observed that IFITM3 was upregulated at both the mRNA and protein levels after *C. parvum* infection. IFITM3 is an innate immune response protein generally known to inhibit the entry and replication of many viruses [55–57]. Individuals lacking IFITM3 are highly susceptible to infection, even when challenged with influenza virus of low pathogenicity [58]. Therefore, IFITM3 is considered as a first line of cell defense in response to viruses. Besides, the protective effect of IFITM3 is also reflected in the long survival of lung resident memory T cells, in which sustained IFITM3 expression facilitated their survival and protection from viral infection during subsequent exposure [59]. In addition, IFITM3 could mediate the autophagic degradation of interferon regulatory factor 3 (IRF3) and negatively regulates tissue-damaging inflammatory production of type I IFNs induced by the virus [60]. Accordingly, we hypothesized that the enhancement of IFITM3 in response to *C. parvum* infection is the result of host immune defense and a self-regulating mechanism to prevent fatal inflammation after *C. parvum* infection; however, the mechanism awaits future exploration.

STAT1 and STAT2 proteins are key mediators of IFN signaling, and are essential components in the cellular antiviral response and adaptive immunity [61]. In various parasite infections, such as those by *T. gondii* and *Schistosoma japonicum*, STAT1 and STAT2 are evoked and mediate the immune response [62,63]. NMI and IFI35 are both IFN-induced proteins: NMI enhances STAT-mediated transcription of downstream genes in the Janus kinase (JAK)-STAT pathway [64]. IFI35 is known to interact with N-Myc and NMI [65]. However, IFI35 has been proven to negatively regulate retinoic acid-inducible gene I protein (RIG-I) antiviral signaling and promote the replication of vesicular stomatitis virus. Therefore, IFI35 might serve as a flexible immunological regulator during pathogen defense [66,67]. In the present study, the upregulation of IFN-induced proteins STAT1, STAT2, NMI, and IFI35 was observed and validated, and the strong interactions in the PPI network further demonstrated the IFN-centered immune reaction against *C. parvum*. The inflammatory molecules mentioned above might be key targets in the anti-*C. parvum* response.

Apart from IFN-centered signaling pathways, the remarkable downregulation of proteins involved in multiple metabolic pathways also reflects *C. parvum*'s dependency on certain nutrients, which also provides candidates for effective drugs or inhibitors. Possessing highly streamlined metabolic pathways, but an inability to *de novo* synthesize nucleosides, fatty acids, and any amino acids, *C. parvum* hijacks and remodels existing host metabolic pathways for its own benefit [10,68]. In the present study, we found that the levels of multiple host metabolic enzymes were downregulated in response to *C. parvum* infection, and then an in-depth exploration was conducted to characterize the proteins' functions. The data revealed widely decreased levels of host metabolic enzymes such as argininosuccinate synthase (ASS1), glutamine synthetase (GLUL), phosphoglycerate mutase 1 (PGAM1), glyoxylate and hydroxypyruvate reductase (GRHPR), NADH: ubiquinone oxidoreductase subunit B6 (NDUFB6), NFS1 cysteine desulfurase (NFS1), pyruvate dehydrogenase E1 component subunit alpha (PDHA1), SDHB, and serine hydroxymethyltransferase 2 (SHMT2) after infection. Folate and thymidylate biosynthesis are of vital importance during DNA replication, and dihydrofolate reductase (DHFR) is a key enzyme in folate metabolism, catalyzing the oxidation of NADPH and reduction of dihydrofolate to NADP and tetrahydrofolate [69,70]. The important roles of folate

metabolism and nucleic acid synthesis in cryptosporidiosis, mean that currently, inosine monophosphate dehydrogenase (IMPDH) and dihydrofolate reductase-thymidylate synthase (DHFR-TS) are common targets for the treatment of cryptosporidiosis [71,72]. However, the parasite has been found to tolerate the loss of these classical targets, possibly via as-yet-undiscovered purine transporters and salvage enzymes. In the present study, the data revealed downregulation of SHMT2 in host cells, which is a pyridoxal phosphate (PLP) binding protein catalyzing the cleavage of serine to glycine, accompanied with the production of 5, 10-methylenetetetrahydrofolate (5, 10-CH$_2$-THF) [73,74]. The downregulation of SHMT2 by *C. parvum* revealed not only obstruction of the host metabolism, but also a potential parasitism strategy of *C. parvum* to sustain an essential nutrient supply. An early study reported the promotion of exogenous purine nucleosides during *C. parvum* infection [75]. However, genomic analysis indicated the loss of pyrimidine *de novo* synthesis in *C. parvum*; therefore, this parasite is almost entirely dependent on import from the host for its purine and pyrimidine requirements [76,77]. SHMT2 is also an essential intermediate for purine biosynthesis; therefore, we concluded that inhibition of SHMT2 caused by *C. parvum* serves as one of the strategies to secure a purine supply from the host. Similarly, we found that ASS1, a urea cycle enzyme that converts nitrogen from ammonia and aspartate to urea, was downregulated after *C. parvum* infection. A recent publication on hepatocellular carcinoma showed that downregulation of ASS1 is associated with a more malignant cancerous phenotype and poor prognosis [78]. In-depth investigation further revealed that ASS1 facilitates pyrimidine synthesis during cancerous proliferation by activating CAD (carbamoyl-phosphate synthase 2, aspartate transcarbamylase, and dihydroorotase complex), through the regulation of aspartate levels [79]. Based on these processes, *C. parvum* would obtain sufficient purines and pyrimidines, which would ensure successful parasitism and replication, possibly by interfering with the host's purine and pyrimidine metabolism.

Besides, we observed significant reductions in the abundances of enzymes involved in the cell oxidative respiratory chain, such as NFS1, PDHA1, and SDHB. Mounting evidence demonstrates the *C. parvum* is an energy-requiring parasite. It is generally accepted that *Cryptosporidium spp.* has lost mitochondria-like organelle-derived energy metabolic capabilities via reductive evolution. Lacking constitutive oxidative phosphorylation, *C. parvum* is unable to complete the tricarboxylic acid (TCA) cycle and cytochrome-based electron transport processes [10,68,80,81]. Thus, gluconeogenesis and glycolysis are the most important energy producing processes in parasites [15,80]. Genomic and biochemical evidence further confirmed the dependence of *Cryptosporidium* on glycolysis as the main energy source and an overall reliance on the host for basic nutrients [10,82–85]. PGAM1 was among the proteins identified in our study, which is an important glycolytic enzyme coordinating glycolysis and biosynthesis, including the pentose phosphate pathway and serine synthesis pathway [86,87]. Evidence showed that some host-parasite homologous genes of the host glycolysis/gluconeogenesis pathways were downregulated, while host-exclusive genes were upregulated during invasion and intracellular development, suggesting parasite-derived competition for metabolic substrates, which might explain the reduction of host enzymes in our study [17]. Another study suggested that sugar transportation occurs between *Cryptosporidium* and host intestine epithelial cells [88]. Parasites further deprive the host cells of the substrates of glycolysis by transportation, and might correspondingly affect the expression of the host genes in the glycolysis pathway. Recent research showed that *C. parvum* could interfere with host glucose transporters (GLUT) 1/2 and Na$^+$-coupled glucose transporter (SGLT) 1 expression, and a significantly higher intracellular glucose level was observed in infected cells, which would point to an adaptation of the host cells' glucose uptake after infection [89]. Based on this, PGAM1 might be a new target for *C. parvum* to inhibit glycolysis and biosynthesis in host cells.

Research has also proven that glutaminolysis and lactate are necessary for parasite replication [15]. In this study, the downregulation of GLUL after *C. parvum* infection would hinder host glutamine synthesis from glutamate and ammonia, and might even promote parasite replication, which is dependent on glutamine catabolism. All these identified enzymes indicate modification of host energy production and conversion by *C. parvum* to sustain its own growth, which might also provide new insights into anti-*Cryptosporidium* targets.

In summary, we observed significant enrichment of interferon-centered signaling pathways and extensive inhibition of metabolism-related enzymes in host cells caused by *C. parvum* infection, providing a deeper understanding of the molecules and their functions involved in the host-*C. parvum* interaction. The identified DEPs and signaling pathways narrowed the range of important functional molecules during anti-parasite immune responses and also confirmed *C. parvum's* deprivation of certain host nutrients, which might be potential targets for *Cryptosporidium* treatment. Moreover, the mechanism by which the identified factors participate in *C. parvum* parasitism and the complex interplay between the identified factors require further investigation.

## Supporting information

**S1 Table. List of significantly regulated proteins in *C. parvum*-infected HCT-8 cells versus uninfected HCT-8 cells identified by LC-MS/MS analysis.**
(XLSX)

**S1 Fig. Basic analysis of host proteins in the mass spectrometry data. (A)** Basic statistics of the mass spectrometry data. **(B)** Length distribution of identified host peptides. **(C)** Mass and sequences coverage ratio of host proteins. **(D)** Mass distribution of identified proteins in host cells.
(TIF)

**S2 Fig. Basic analysis of identified *C. parvum* proteins in the mass spectrometry data. (A)** Basic statistics of the mass spectrometry data of identified *C. parvum* proteins. **(B)** Length distribution of identified *C. parvum* derived-peptides from host cells. **(C)** Mass distribution of identified C. parvum proteins from host cells. **(D)** Distribution of the number of peptides per protein.
(TIF)

**S3 Fig. Enrichment and distribution of host DEPs presented in a bubble pattern according to their GO functional classification.** GO enrichment analysis of upregulated (A) and downregulated proteins (B) in the Cellular Component category. The size of the circle area represents the number of DEPs, and the color represents the *P* value of the enrichment significance of the DEPs under the GO classification. The redder and more distributed to the right the circles, the more important their classification.
(TIF)

**S4 Fig. Bioinformatic analysis of identified *C. parvum* proteins from infected host cells. (A)** COG/KOG analysis of identified *C. parvum* proteins. **(B)** GO enrichment analysis of parasite-derived proteins. The size of the circle represents the number of DEPs in that functional class or pathway, and the color represents the significance of the enrichment (*P* value). **(C)** GO enrichment analysis of identified *C. parvum* proteins in the Biological Process category. **(D)** *C. parvum*-derived proteins that were enriched significantly in the ribosome pathway from the KEGG pathway analysis. Protein names in red are parasite-derived proteins identified from

infected HCT-8 cells.
(TIF)

**S5 Fig. Protein-protein interaction network of identified *C. parvum* proteins generated using Cytoscape.** The degree represents the strength of the protein interaction.
(TIF)

## Acknowledgments

We thank Jingjie PTM Bio (Hangzhou, China) CO., LTD for technical support.

## Author Contributions

**Conceptualization:** Teng Li, Yujuan Shen, Jianping Cao.

**Formal analysis:** Teng Li, Yujuan Shen, Jianping Cao.

**Funding acquisition:** Hua Liu, Yujuan Shen, Jianping Cao.

**Methodology:** Teng Li, Hua Liu, Nan Jiang, Yiluo Wang, Ying Wang, Jing Zhang.

**Supervision:** Yujuan Shen, Jianping Cao.

**Writing – original draft:** Teng Li.

**Writing – review & editing:** Jianping Cao.

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
