## [Decision Letter · Decision Letter 0]

7 Sep 2021

Dear Prof. Cao,

Thank you very much for submitting your manuscript "Comparative proteomics reveals Cryptosporidium parvum manipulation of host cell molecular expression and immune response" for consideration at PLOS Neglected Tropical Diseases. As with all papers reviewed by the journal, your manuscript was reviewed by members of the editorial board and by several independent reviewers. The reviewers appreciated the attention to an important topic. Based on the reviews, we are likely to accept this manuscript for publication, providing that you modify the manuscript according to the review recommendations. 

Sincerely,

Hamed Kalani

Associate Editor

Fabiano Oliveira

Deputy Editor

Reviewer's Responses to Questions

**Key Review Criteria Required for Acceptance?**

**Methods**

-Are the objectives of the study clearly articulated with a clear testable hypothesis stated?

-Is the study design appropriate to address the stated objectives?

-Is the population clearly described and appropriate for the hypothesis being tested?

-Is the sample size sufficient to ensure adequate power to address the hypothesis being tested?

-Were correct statistical analysis used to support conclusions?

-Are there concerns about ethical or regulatory requirements being met?

Reviewer #1: THIS IS A WELL CRAFTED STUDY, AND UNQIUE LOOK AT C. parvum's AFFECT ON THE HOST. THEY UTILIZE HCT-8 cells AS A MODEL FOR ISC, AND THE STUDY LOOKS A DEP's AND HOW THIS AFFECTS THE HOST CELL. IT IS AN EXCELLENT STUDY WITH EXCELLENT CONTROLS TO PROVE THEIR DATA. THE DATE WAS ANALYZED BY SEVERAL MEANS AND I FOUND THEIR DATA AND ANALYSIS QUITE CONPELLING.

Reviewer #2: Methods fully explained

**Results**

-Does the analysis presented match the analysis plan?

-Are the results clearly and completely presented?

-Are the figures (Tables, Images) of sufficient quality for clarity?

Reviewer #2: Yes the results match the plan. But the mass of results ultimately leads to a manuscript that reads like a table rather than a concise and focused manuscript. 

**Conclusions**

-Are the conclusions supported by the data presented?

-Are the limitations of analysis clearly described?

-Do the authors discuss how these data can be helpful to advance our understanding of the topic under study?

-Is public health relevance addressed?

Reviewer #2: There are lots of results, but no real conclusion: from the abstract "This systematic analysis of the proteomic of C parvum-infected HCT-8 cell identified a wide range of functional proteins that participate in host anti-parasite immunity or act as potential targets during infection, providing a new insight into the molecular mechanism of C parvum infection. "

**Editorial and Data Presentation Modifications?**

Reviewer #1: (No Response)

Reviewer #2: Fine overll. There are singular words that should be plural and plural words hat should be singular.

**Summary and General Comments**

Reviewer #2: The strength of the work is the work that was done. The weakness is that it reads like a list in the methods and in the discussion. However, greet data that should be shared.

PLOS authors have the option to publish the peer review history of their article (what does this mean?). If published, this will include your full peer review and any attached files.

Reviewer #1: Yes

Reviewer #2: No

Figure Files:

Data Requirements:

Reproducibility:

References

---

## [Editor Report · Decision Letter 1]

5 Oct 2021

Dear Prof. Cao,

Thank you very much for submitting your manuscript "Comparative proteomics reveals Cryptosporidium parvum manipulation of host cell molecular expression and immune response" for consideration at PLOS Neglected Tropical Diseases. As with all papers reviewed by the journal, your manuscript was reviewed by members of the editorial board and by several independent reviewers. The reviewers appreciated the attention to an important topic. Based on the reviews, we are likely to accept this manuscript for publication, providing that you modify the manuscript according to the review recommendations. 

Dear Authors,

Your manuscript was read by several reviewers and according to the opinion of the reviewers, your manuscript needs to be revised. Please pay attention to the suggestions provided below.

1- The results match the plan but the mass of results ultimately lead to a manuscript that reads like a table rather than a concise and focused manuscript.

2- There are singular words that should be plural and plural words that should be singular.

3- The weakness is that the manuscript reads like a list in the methods and in the discussion and should be revised.

Sincerely,

Hamed Kalani

Associate Editor

Fabiano Oliveira

Deputy Editor

Figure Files:

Data Requirements:

Reproducibility:

References

---

## [Editor Report · Decision Letter 2]

25 Oct 2021

Dear Prof. Cao,

We are pleased to inform you that your manuscript 'Comparative proteomics reveals Cryptosporidium parvum manipulation of the host cell molecular expression and immune response' has been provisionally accepted for publication in PLOS Neglected Tropical Diseases.

Best regards,

Hamed Kalani

Associate Editor

Fabiano Oliveira

Deputy Editor

---

## [Editor Report · Acceptance letter]

10 Nov 2021

Dear Prof. Cao,

We are delighted to inform you that your manuscript, "Comparative proteomics reveals Cryptosporidium parvum manipulation of the host cell molecular expression and immune response," has been formally accepted for publication in PLOS Neglected Tropical Diseases.

Best regards,

Shaden Kamhawi

co-Editor-in-Chief

Paul Brindley

co-Editor-in-Chief
